# Indigenous Peoples' Perceptions of Their Food System in the Context of Climate Change: A Case Study of Shawi Men in the Peruvian Amazon

Ingrid Arotoma-Rojas [1,*] , Lea Berrang-Ford [1], Carol Zavaleta-Cortijo [2,3], James D. Ford [1] and Paul Cooke [4]

1   Priestley International Centre for Climate, University of Leeds, Leeds LS2 9JT, UK
2   Unidad de Ciudadanía Intercultural y Salud Indígena (UCISI), Facultad de Salud Pública y Administración, Universidad Peruana Cayetano Heredia, Lima 15074, Peru
3   Nutritional Epidemiology Group, School of Food Science and Nutrition, University of Leeds, Leeds LS2 9JT, UK
4   School of Languages, Cultures, and Societies, University of Leeds, Leeds LS2 9JT, UK
*   Correspondence: eeiear@leeds.ac.uk

**Abstract:** Biodiversity and ecosystem conservation in the Amazon play a critical role in climate-change mitigation. However, institutional responses have had conflicted and complex relations with Indigenous peoples. There is a growing need for meaningful engagement with—and recognition of—the centrality of Indigenous peoples' perceptions and understanding of the changes they are experiencing to inform successful and effective place-based adaptation strategies. To fill this gap, this study focuses on the value-based perspectives and pragmatic decision-making of Shawi Indigenous men in the Peruvian Amazon. We are specifically interested in their perceptions of how their food system is changing, why it is changing, its consequences, and how/whether they are coping with and responding to this change. Our results highlight that Shawi men's agency and conscious envisioning of their future food system intersect with the effects of government policy. Shawi men perceive that the main driver of their food-system changes, i.e., less forest food, is self-driven population growth, leading to emotions of guilt and shame. During our study, they articulated a conscious belief that future generations must transition from forest-based to agricultural foods, emphasising education as central to this transition. Additionally, results suggest that the Peruvian government is indirectly promoting Shawi population growth through policies linking population size to improved service delivery, particularly education. Despite intentional Shawi moves to transition to agriculture, this results in a loss of men's cultural identity and has mental-health implications, creating new vulnerabilities due to increasing climatic extremes, such as flooding and higher temperatures.

**Keywords:** food system; indigenous; climate change; food security

## 1. Introduction

Food systems are major contributors to, and are being impacted by, climate change, with implications for people's physical and mental health [1–4]. Approximately one-third of greenhouse emissions are attributable to food systems [1,2,5], reaffirming their centrality in current efforts to mitigate climate change [2,6]. At the same time, climate change is already impacting food systems. For example, food production, food supply chains, and food consumption are frequently affected by extreme events that negatively impact food quantity, prices, and quality [1,7,8]. These impacts are increasing malnutrition and the risk of food insecurity and hunger, and climate-driven ecosystem and biodiversity loss are also affecting mental health [1,2,6].

Climate-change risks are experienced differentially across the globe, disproportionally affecting people with already-vulnerable food systems, especially Indigenous peoples [1–3,9–11]. Indigenous peoples living in biodiversity hotspots like the Amazon have been particularly

affected as their access to land has been compromised. This has been compounded by globalisation, ongoing colonisation, and environmental disruptions, causing rapid transitions in their food systems and resulting in modifications to diets, nutritional status, and physical activity [9,12–14]. This food transition is characterised by an increased dependence on external market foods, combined with decreased availability of food from the forest, with consequences for Indigenous peoples' well-being, health, nutrition, culture, and social networks [15–18]. Ongoing development underpins key drivers of this food transition, overlapping substantially with climate-change drivers, including deforestation, urbanisation, road construction, increasing agro-exportation, and pollution due to extractive industries that contribute to ecosystem and biodiversity loss [12,19,20].

Institutional responses focused on climate and food in the Amazon have had conflicted and complex relations with Indigenous peoples. In Peru, for example, government food-aid programs to reduce undernutrition have focused on supplying highly processed foods, which have been found to conflict with traditional food preferences; research found limited evidence of improvements in nutritional indicators among Indigenous peoples as a result of these programs, largely due to their misalignment with Indigenous cultures and customs [21,22]. Additionally, climate-change mitigation and conservation policies have made the Peruvian government increase the number of protected areas, limiting the access of Indigenous peoples to forest resources and aggravating Indigenous land dispossession [23,24].

Successful and effective place-based adaptation strategies to overcome food insecurity in a changing climate require meaningful engagement with—and recognition of—the centrality of Indigenous peoples' perceptions of the changes that they are experiencing [25,26]. Consideration of Indigenous peoples' perceptions, emotions, knowledge systems and worldviews, and the implications of all these factors for our understanding of climate-change impacts in food systems on biodiversity hotspots like the Amazon, is limited [27,28]. There is an urgent need for Indigenous peoples' voices and values to be taken into consideration and to recognise the emotional and psychological impact that climate change has on social and cultural values and ways of living [29–32].

To fill this gap, this research aims to identify, characterise, and assess Indigenous peoples' value-based perspectives on current and potential adaptation strategies and trade-offs related to food systems in a changing climate in the Amazon. We are specifically interested in Indigenous peoples' perceptions of how their food system is changing, why is it changing, what the consequences are of this change, and how/whether they are coping and responding effectively to it. Taking as our starting point their perceptions, we assess individual, communal, and societal determinants interacting with Indigenous food systems, following a collective health framework. This is done in collaboration with the Shawi Indigenous men of Nuevo Progreso, a community in the Peruvian Amazon.

## 2. Materials and Methods

### 2.1. Conceptual Framework

In this research, we first start with the perceptions of individual people about the changes they are currently experiencing in their food system in comparison with the past and their envisioned futures, seeking to identify what is worth preserving and achieving in the interaction with climate change and ecosystem changes [29,33]. By placing perceptions at the centre of our analysis, we seek to identify how people live and experience the world [34], privileging values that are subject to intangible or non-economic harm (e.g., sense of place, knowledge, dignity, self-determination) [30], as well as emotions affecting mental health and well-being related to climate change [35,36]. We do so in response to the recognition that Indigenous peoples' mental health is particularly affected by climate change, with well-documented evidence of ecological grief associated with current and future ecological, cultural, and identity loss [31,32,37]. Despite this emerging and growing literature, however, there are limited data on Amazonian Indigenous peoples [28].

Second, our approach is underpinned by the understanding that the foods people eat are determined and influenced by individual choices, rooted in family traditions and livelihoods, which in turn are embedded in societal and cultural contexts that condition an individual's relative autonomy [38–40]. Based on the collective health framework, this study is predicated on the understanding that there is a dialectical relationship between individuals and society structured according to three analytic dimensions: (a) societal: the socio-historical and political system at a local, national, and global level; (b) community: ways of living life embedded in social groups, cultural identities, and economic contexts; and (c) individual: the lifestyles and behaviours of individuals and the individual psycho-emotional conditions that influence health [39,40].

Finally, we additionally analyse how climate interacts with the three analytical dimensions that influence food systems (Figure 1) [3]. Vulnerability to climate change is assumed within this framework to be embedded and subsumed within complex and dynamic socio-ecological factors interacting across different spatial-temporal scales that put people at risk: where, how, when, and why certain groups are more vulnerable than others [11,41].

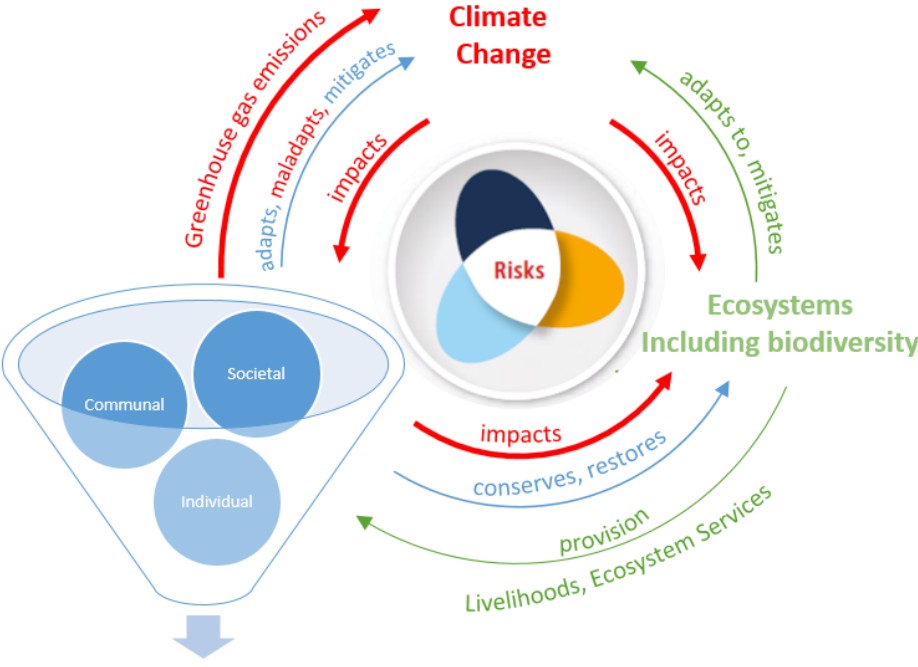

**Figure 1.** Conceptual framework of food systems and climate change. Source adopted from Pörtner et al. (2022) [1] and modified by the authors.

### 2.2. Study Area

We used a case-study approach to develop an in-depth and place-based understanding about the complexity of vulnerability and adaptation to climate change [42]. This research was possible due to an ongoing relationship between the Indigenous Health Adaptation to Climate Change Program "www.ihacc.ca (accessed on 7 December 2022)" and the Shawi people of the Peruvian Amazon that has lasted over 10 years. Our focus on the intersection of food, health, and climate was previously identified by community members as being one of the most pressing issues they face, and one that needs immediate attention in policy and research agendas [43–45]. Based on a household questionnaire in 2014, a study found that food insecurity and malnutrition levels among the Shawi were among the highest globally: 98% of Shawi households were food insecure, 66% of children had anaemia, 44% were stunted, 17% were underweight, and 19% of anaemic children had an overweight parent [22]. The aim of the study described in this paper is to complement current ongoing research with a deeper exploration of Indigenous peoples' narratives and perceptions of

the changes experienced in their food system, reframing the research to centre Indigenous people's own voices more prominently alongside scientific research.

This paper focuses on a Shawi Indigenous community in the north-western Peruvian Amazon, Nuevo Progreso (Figure 2). This community is formed of three localities: Nuevo Progreso, Nuevo Yurimaguas, and Nuevo Belén. At the time of research in 2019, there were approximately 162 households (Nuevo Progreso: 97, Nuevo Belén: 50, and Nuevo Yurimaguas: 56). Nuevo Progreso did not have electricity, but Nuevo Belén and Nuevo Yurimaguas had solar energy. None of the communities had access to potable water. In 2010 the Peruvian government built the first road connecting Nuevo Progreso with the largest nearby city, Yurimaguas.

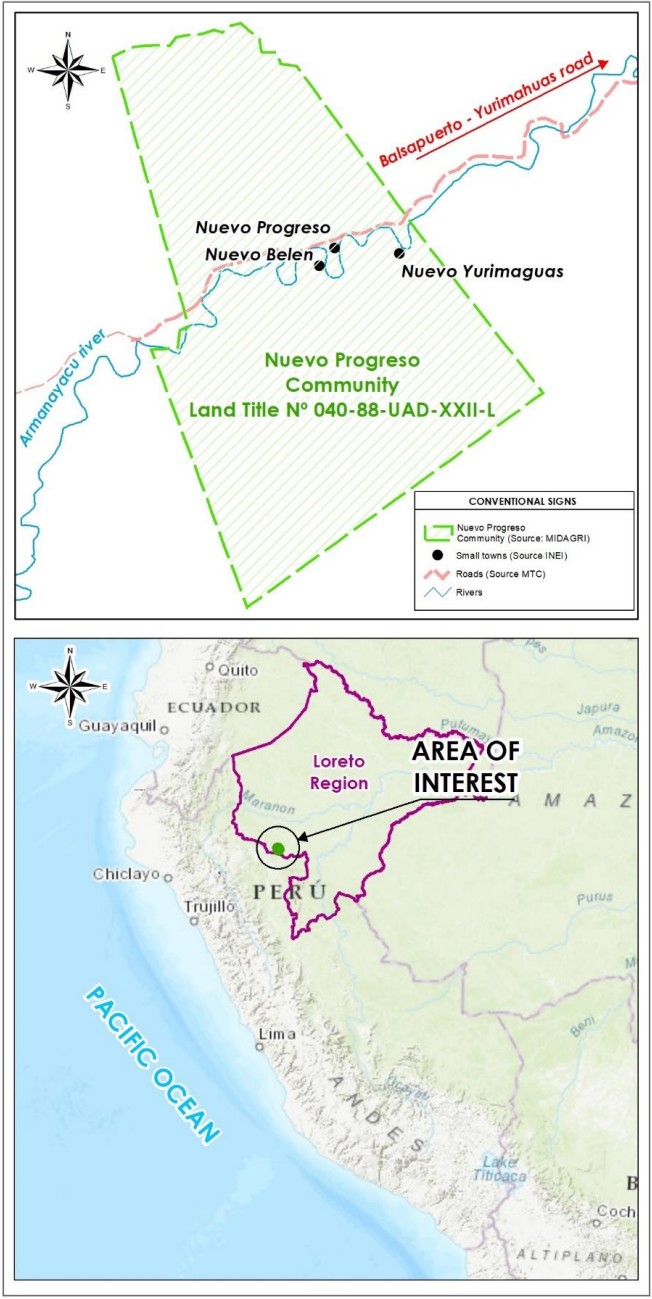

**Figure 2.** Map outlining the study area made by Paul Castro for this manuscript with ArcGis 10.8.1 software. Source: official database from the Ministry of Agriculture of the Peruvian Government (community limits), National Institute of Statistics and Informatics (town locations), Ministry of Transport and Communications (roads).

The Shawi people comprise more than 20,000 people living in 185 communities in the Loreto and San Martín regions [46]. As a result of the extended presence of Jesuits (1638–1768) and Christian missionaries (since 1945), there is Christian–Shawi worldview syncretism in their understanding of everyday life [47–49]. Gonzales-Saavedra [47] noted, however, that the Shawi transformed the belief that they were among the easiest Indigenous peoples to be Christianised into a survival strategy, designed to preserve their identity, culture, and traditional practices in colonial and postcolonial times.

Spiritualism is a cornerstone of the Shawi worldview and is key to understanding their relationship with nature [48,50,51]. Traditionally, the Shawi understand the world as being "round as a honeycomb wasp" with nine specific spaces ruled by sentient spirits (Figure 3). The third space is Nu´paru´te, where the Shawi people live in harmony with plants, animals, and the forest's spirits [50]. The Shawi understand that their relationship with every spirit (mountains, trees, plants, etc.) must be in balance to live well; diseases and ecological changes are, therefore, sometimes understood as a punishment for breaking this balance [47,48,50,52,53].

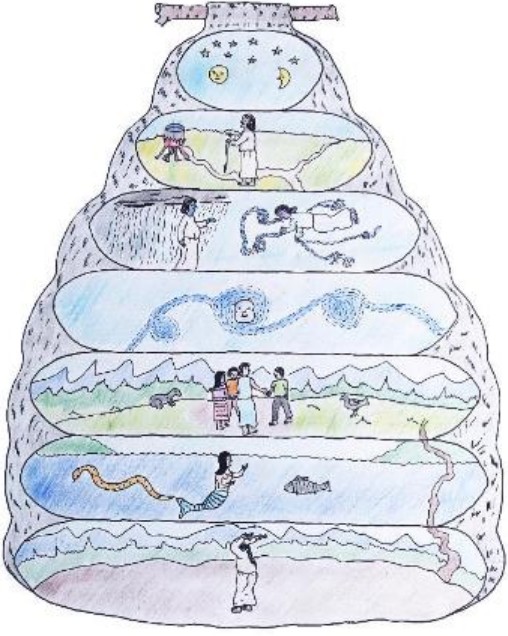

(8) Yuhkiru´te, the space of the moon; and (9) Pi´iru´te, the space of the sun where the Shawi deity Kumpanamá lives.

(7) Chimirinru´te Nanpirinru´te, the space of the dead;

(5) U´nanru´te, the space of rain; (6) Iwanru´te, the space of wind;

(4) the space of air where only the spirits of the Shamans go to heal or damage people;

(3) Nu´paru´te, the space where the Shawi people live in interrelation with the plants and animals, as well as with the forest's spirits;

(2) Iiru´te, the space where all the animals and spirits of water live;

(1) Ahkupuru'te or space inside the earth, ruled by Uhkuá who holds the earth in its hand, which sometimes trembles when it gets tired, making the earth shake

**Figure 3.** Shawi worldview, drawn by Alberto Chanchari from the Shawi people of Nuevo Progreso for this manuscript.

### 2.3. Methods

Fieldwork was conducted in Nuevo Progreso in January and February 2019 by the lead author (I.A.-R.), accompanied by a young Shawi man designated by the community as a gatekeeper following customary law. Research characteristics were debated and approved in a community assembly. Following the assembly, each participant had the right to decide whether to participate. By request of the community assembly, the community's name was not anonymised as a strategy to promote their historical records for future generations.

All interviewees were adults, heads of their families, and active participants in community assemblies. Our focus on men was not deliberate but rather emerged from the dynamics of community gender roles. In many Indigenous cultures in the Amazon like the Shawi people, women are not allowed to participate in public spaces, and they are less likely to attend schools or to speak Spanish in comparison to men [54].

We undertook 23 semi-structured, in-depth interviews with Shawi men, 5 participant-observation activities, and a male-only focus group. Interviews were conducted by I.A.-R. primarily in Spanish, as preferred by participants, with translator assistance when questions

were unclear. Interviews were primarily located in Nuevo Progreso ($n = 14$), with some in Nuevo Belén ($n = 4$) and Nuevo Yurimaguas ($n = 2$). In Yurimaguas, the closest city, we interviewed the community's school representative and two representatives of the Shawi Indigenous organisation ($n = 3$). Only three interviews were not recorded, respecting the interviewees' decision.

The Interviews covered issues related to food and health changes. First, we asked about food consumption on the day of the interview and the week before. Men were asked to compare their current diet with their childhood diet. We also explored Indigenous knowledge to discuss important and dominant foods consumed and how access and consumption had changed over time. We asked participants to explore and characterise their perceptions of the drivers of these changes, considering the three analytical dimensions of the conceptual framework: individual, collective, and societal drivers of change. When weather variability was not mentioned, we probed further to inquire whether/how weather interacts with the changes they are experiencing around food. The last part of the interview focused on visions of the future, both desired and foreseeable. Regarding their desired future, we asked participants to choose three things they would change/improve in their community if they had the power to and to explain their reasoning. In the case of the foreseeable future, we asked them to describe what the community might look like in 5 and 10 years if things continued without intervention. If food was not mentioned, we asked directly about what they might eat in that future and how activities around food were perceived to have evolved. In many cases, discussions around food led to a broader discussion on well-being and health. These discussions also included a focus on spiritual dimensions.

I.A.-R. joined and observed food- and health-related activities with community members, including participation in fishing and gathering fruits from the forest, family farming and crop production, cooking wild food from hunting, preparing Masato (a local beverage), and preparing plant-based medicines. During the field research, a diary was kept to document activities, feelings, and observations, which served as an additional data source for our research findings.

Following the initial phase of interviews, a focus group was conducted with men ($n = 12$) to collectively discuss overall results. We first asked participants to identify food-related activities, and then to explain how they had changed over time and how they thought these activities would evolve. Additionally, we explored characteristics of Indigenous knowledge by asking them about abilities that they felt the Shawi people have that other people do not. We also explored their approaches to seeking medical attention.

All recorded interviews and group discussions were transcribed verbatim and hand-checked for accuracy. This process also included making a note of the speaker's emotions, something that can be lost in audio transcription. Here we followed the principles of thematic analysis [55]. For this research, inductive codes (data-driven) were created (e.g., changes in hunting), which were then organised into broad theory-driven families of code to sort findings (e.g., food-system changes, climate-change perceptions, envisioned futures). The coding was completed using nVivo data-analysis software. All results met data saturation (i.e., were reported by all or most respondents) or were otherwise reported explicitly as divergent or outlier findings.

### 2.4. Positionality

The fieldwork and primary analysis were conducted by I.A.-R., a Quechua female based at a UK university, with the help of a young Shawi male research assistant. We acknowledge that the work's focus on men may have biased participant responses and researcher interpretations of the results. This research was intentionally framed to focus on the voices of Indigenous Shawi men to minimise bias from researcher interpretation. Our results, therefore, are primarily descriptive, aimed at centring the voice of Shawi men with limited analytical interpretation or filtering. Complementing this, in the discussion

we undertake a more interpretive and researcher-driven analysis consistent with scientific axioms and epistemology.

## 3. Results

### 3.1. Individual Dimension

All interviewers highlighted that the Shawi food system is based on three sub-systems, composed of wild food from the forest, sourced through hunting, fishing, and gathering; food cultivated through farming; and food acquired from external sources, from shopping or government aid (Figure 4). Food scarcity has become a significant challenge in everyday life for this group and is directly associated with reductions in wild foods available from the forest. Most interviewees explained that they usually eat only once a day. To cope with hunger, the Shawi constantly drink Masato, a fermented cassava drink that even small children consume, which makes them "feel full".

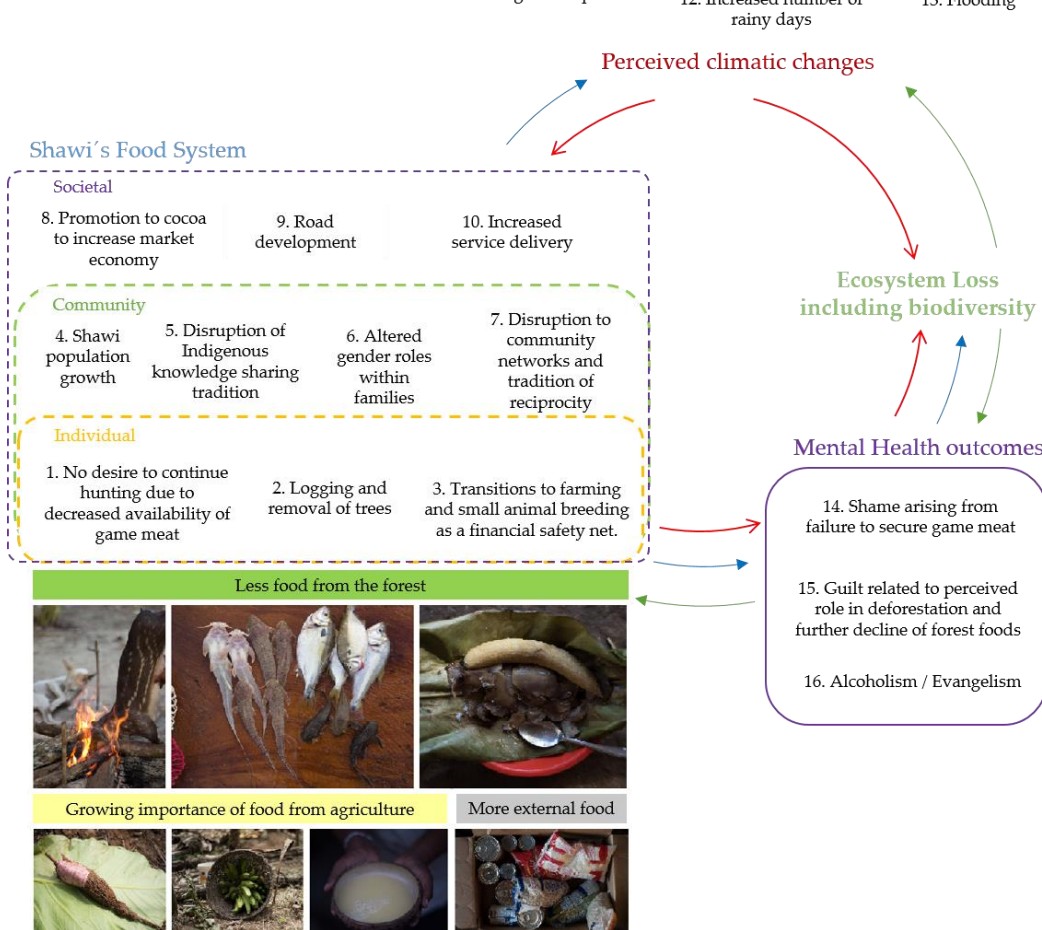

**Figure 4.** Shawi men's narratives of changes of their food system. Source: pictures taken by Matthew King for the IHACC program.

Forest foods are still the most preferred but also the scarcest; fish and game meat are highly preferred, although they are largely acquired from external sources. Food from farming, such as cassava or plantain, is the most readily available, but preference for these foods is declining since they are often the only thing people eat for many days. Hens have also been identified as a preferred food. Other domestic animals and crops like cocoa are less preferred among the Shawi and are considered, rather, as a means to obtain food from the market.

*"Our women tell us to go find something to feed our kids. ( . . . ) sometimes when I go hunting, I cannot find anything, I even go further in the forest and nothing! Then I have to come back late, and I feel upset . . . You can no longer find anything here." –Shawi adult.*

Many men recalled that the last time they had hunted large animals such as deer or collared peccary was several years before. Nowadays, their perception is that the difficulties associated with hunting do not depend on their skills but on luck. Most men explained that they no longer hunt because it requires dedicating whole days to going far into the forest, and even then it is difficult to find animals. Many men prefer not to go hunting to avoid the potential embarrassment of returning home without any prey (1 in Figure 4). The inability to hunt regularly and effectively affects Shawi men's well-being.

Nowadays, men tend to carry a shotgun in case they find small wild animals such as partridge (*Cripturellus cinereus*) while farming. However, hunting requires cartridges only available in the nearest city, generating a dependency on money, and making it even more challenging to continue hunting. During research visits in Nuevo Belén, one of the interviewees captured a small peccary but reported that it was only by luck because, although he did not have a cartridge and so could not shoot it, the small bird was with its mother and he was able to grab the baby.

With less availability of game meat, the Shawi diet relies on fish (fresh or salted) as the main protein source. However, interviewees noted that fish is no longer obtained from the river but is increasingly acquired from external sources. All participants explained that there are only small fish available in their territory, so they prefer to buy from vendors on the road from the city of Yurimaguas. External purchasing of fish could reflect a strategy to recover local fish stocks but was mainly reported to reflect resignation in the face of meagre fishing in local waters. Money to buy fish came from selling domestic animals, crops, or seasonal wild fruits.

Given the absence of game meat and fish, food predominantly coming from the forest included wild fruits, mushrooms, edible plants, and small animals like suri, a worm that lives in the aguaje or palmito trees. However, many strategies for collecting forest food involve cutting down trees, which further reduces forest-food availability (2 in Figure 4). For example, to obtain aguaje fruit, men must cut down an entire tree. This fruit is used to complement meals and has recently become an important source of seasonal income. Thus, whenever a family needs protein, they go to the forest where an aguaje tree has fallen, peel a part of the stem, and remove the suri worms. Given the increased need for cash to purchase external foods, many aguaje trees have been cut down, leading to worries about the future availability of the aguaje fruit and suri worms.

*"We have quantity in the forest, it is our natural farm, there is a lot of wealth there! (...) now there is less because sometimes people are destroying a lot to sell at the market. Our people are doing it, and it is no longer like before, ( . . . ) they chop the trees down, and they are killing them. For it to grow up, how many years? ( . . . ). Our next generation, what are they going to eat now?" –Shawi adult.*

Due to the decline of forest food, Shawi men are changing their livelihood activities. All participants recognised that because the forest has changed so rapidly, they now rely more heavily on farming and domestic animals for food (3 in Figure 4). Farming combines subsistence crops with cash crops, such as cocoa, to generate income. Subsistence crops are traditionally cassava and plantain and are available all year. These tubers are sometimes sold when a harvest is abundant. It is now easier to travel to Yurimaguas given the recent establishment of a road to the city, and men reported selling crops at the market and using the income to purchase other items. Men also reported growing corn for sale and feeding hens. Bred domestic animals, such as hens, pigs, cows, guinea pigs, and sheep, are perceived as important complements to the Shawi diet. Particularly important for the men was the fact that this allows them to add eggs to their diet, albeit in low quantities. Additionally, selling domestic animals was considered a safety net for emergencies when cash is needed to pay for food, school supplies, or medical care.

Many men perceived that the solution to food scarcity is tied to learning new ways to produce diverse crops, building fish farms to have reliable access to this food, and improving their skills in breeding domestic animals. They reported high expectations from formal schooling to supply this knowledge to younger generations, thereby promoting more reliable access to food.

> *"I told my son: 'you must work, you must make a fish farm so that you can feed your family'. I told him that, and asked him to raise chickens, pigs ( . . . ) There are no more animals in the forest, no more fish either. The forest will not recover. New generations will not know how to find food in the forest ( . . . ) I told them to raise domestic animals so they could eat. I raise my children telling them that."* –Shawi adult.

Logging in the Shawi community also affected forest-food availability (2 in Figure 4). Men explained that they allowed external loggers to enter the community as a source of income to purchase food and school supplies and pay for medical costs or transportation to the nearest city. They recognised that logging contributes to "making animals flee" due to noise and disruptions to their natural habitat, making it harder to hunt game meat. However, many Shawi men still depend on logging, as it provides a substantial income to feed their families, exceeding the amount obtained by selling farm animals or wild fruits in small quantities at the market.

> *"On Sunday I had a little money; the loggers paid me 200 PEN. And I said, 'I am going to buy soap'; we do not have soap and I walked with my little daughter. So I bought my soap and there was a little money left. Then I bought a kilo of collared peccary; 25 soles buys a kilo of collared peccary in Yurimaguas ( . . . ). Nowadays this is a luxury we can only afford a few times."* –Shawi adult.

*3.2. Community Dimension*

All the participants reported that they believe the main reason for a decrease in forest-food availability is their own population growth, articulating that there are now more people to feed. This, they suggested, is resulting in higher demand on forest resources, which in turn incentivises them to take logging deals for cash income to supplement diets as well as cut down trees to sell fruits, all of which are combining to exacerbate a decline in forest access and wild-food availability (4 in Figure 4). This reinforces their perception that they are the cause of the changes they are experiencing and are active agents driving the decline in food from the forest. While assuming responsibility as primary drivers of changes in the availability of forest food, Shawi men also articulated a desire for their community to become more urbanised, supporting continued population growth. The rationale behind this was linked to the expectation of population growth leading to improved access to government services in their communities, such as electricity, drinking water, health services, and education. However, they recognised that this would also mean the continued decline or disappearance of the forest-food sub-system.

> *"I want the population to grow. There will be more people, more young people, more children. ( . . . ) There will be less game meat, but what can we do? ( . . . ) There is not going to be anything anymore ( . . . ). This little girl would not know the taste of agouti or deer . . . . This is what is happening now. My son has eaten deer and collared peccary, but this little girl has never, and she won't."* –Shawi adult.

The decline of forest food plays a central role in the transmission of traditional knowledge (e.g., hunting and fishing techniques) from Shawi men to younger generations (5 in Figure 4). Male disengagement from hunting is disrupting the knowledge sharing that the Shawi have preserved as a critical element of their cultural heritage, in which children would go with the family's men to hunt and learn to differentiate animal tracks, make hideouts, and shoot. Additionally, because hunting now requires more time, it conflicts with school attendance, making it more challenging to continue the intergenerational transmission of knowledge. During research visits in Nuevo Progreso, for example, we observed two boys going out with a slingshot every day to hunt birds. Only once did we observe

the smallest of the boys catching a bird, allowing him to provide breakfast for his family (Figure 5). It was a very proud moment for the family, as they explained that the only knowledge commonly transmitted to younger generations today is how to use a slingshot.

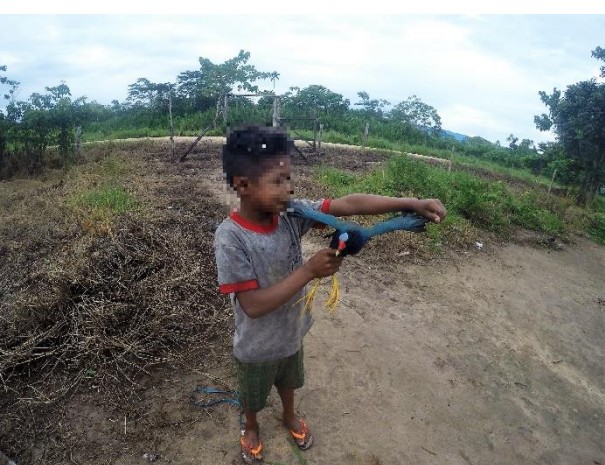

**Figure 5.** The day A (9 years old) proudly caught a bird with his slingshot to feed his family. Source: picture taken by I.A.-R. and shared with the permission of A and his parents.

Second, the reduced availability of forest food that has led to male disengagement with hunting has also altered gender roles within Shawi families (6 in Figure 4). Changes and disruptions in the availability of forest food have meant that men can no longer fulfil their traditional role as hunters, leading many of them to transition to market-related activities (farming and logging) to provide food for their families.

Third, the declining availability of food from the forest was reported to be affecting community networks traditionally based on reciprocity through food sharing (7 in Figure 4). Nowadays, since there is negligible game meat and the population has grown, it is difficult to share food with others, altering the traditional community customs of caring for vulnerable people, particularly elders. For example, during our visit, some interviewees reported the recent death of an older woman whose husband had died and who was dependent on her children for food. With less food in the community, her children could not share much because they had their own families to feed. The older woman presented general discomfort or malaise and did not have enough energy to leave her bed or eat. Some men believed that the woman died because of "revenge disease" or "damage," a disease caused by the envy of others; however, one person suggested that it was hunger and not "damage" that killed this person.

### 3.3. Societal Dimension

The Peruvian government has played a role in the food-system changes that the Shawi men are experiencing. In farming, cocoa cultivation has been extended in the communities because it is part of the regional government's plan to improve connections to the market economy (8 in Figure 4). Interviewees made it clear that families received free seeds, ploughing tools, and technical support. The cocoa project was advertised in the Shawi communities as a way to overcome poverty and promote economic inclusion. Many Shawi men reported that this program was unsuccessful since they were unable to obtain a fair income to cover their monetary needs to buy food and other items. They specified that chocolate factories in the region only buy cocoa in large amounts, which requires Shawi men to sell cocoa to intermediaries at a cheaper price.

Additionally, changes in the forest have expanded and increased in recent years because of logging, facilitated by the completion of a road connecting Balsapuerto to Yurimaguas in 2016, constructed by the Peruvian government to promote road connectivity in the Amazon (9 in Figure 4). Many Shawi men were pleased with the road because it made

selling crops in the market more accessible and created opportunities for tourism. It was unclear whether they associated road construction with impacts on the forest; there was, however, general resignation that they had very little agency in influencing development trends. Along with logging, another dominant activity in the area attributed to the road was the development of papaya plantations. Interviewees mentioned that when the road was built, some men had rented their land to papaya plantations. However, the plantations moved further into the forest in search of more adequate land and so the community soon lost this income stream. During our visit, papaya plantations were located near Balsapuerto. Logging and papaya plantations were perceived to be compounding changes in forest cover, affecting the availability of game meat for the Shawi.

Despite a push by the government for family planning [56], our research also showed that population growth is indirectly encouraged by the Peruvian government. For Shawi men, population growth is perceived as the only way to mobilise governmental provision of social services to the community, such as electricity, clean water, and schools (10 in Figure 4). For example, during our visit, authorities from Nuevo Yurimaguas reported that they had persistently requested that provincial authorities provide an elementary school in the community, as children struggled to cross the river alone to attend the closest school. They reported that the authorities based their decision on the number of local pupils, and Nuevo Yurimaguas did not meet the required minimum, so they were denied this provision. In this sense, population growth represents a key strategy among the Shawi to increase their influence on the government's service planning and provision.

### 3.4. Perceived Climatic Changes

Climatic conditions and changing weather exacerbate Shawi food-system vulnerabilities by affecting food obtained from the farming and breeding of small animals. Shawi men observed that temperatures are increasing, impacting their regular activities (11 in Figure 4). Higher temperatures, for example, reduce the time men can dedicate to farming because of the absence of cool zones to seek shelter and rest. Temperatures are also changing the timing of farming activities, making it preferable to work early in the morning when the heat is considered more bearable. Furthermore, Shawi men reported that increasing air temperatures are associated with increasing water temperatures in rivers and streams. Some men reported that the increase in water temperatures also affects the availability of fish.

Many men reported observing precipitation changes, with an increase in the number of rainy days in winter (12 in Figure 4). When it rains, they do not go out to hunt and prefer not to farm, since they feel that they could get sick. Additionally, rain increases streamflow, which, they reported, makes it more challenging to fish and difficult for them to manoeuvre canoes, since the river can rise rapidly under these conditions. A month before our visit, a woman and her small children died trying to cross the river at night with a canoe after heavy rain.

Shawi men recognised that since 2014, flooding had been a recurrent event affecting them every two years (13 in Figure 4). Many men explained that flooding mainly affects crops and small animals like hens, generally considered their safety net to cope with rapid changes in their food systems. Flooding in the study area can last several hours, damaging food sources. Many men recalled losing their breeding animals, especially hens, and losing corn and cassava crops. Flooding has motivated Shawi men to identify the most at-risk places in the community so they can avoid planting in those areas. Nonetheless, higher-altitude areas are scarce, and only some areas are considered suitable for relocation and farming.

> "When it floods, sometimes it takes our things ( ... ). In the first flood, it was 2014, that time was at dawn, 1 am, so it was flooded when we were sleeping, so many chickens were taken." –Shawi adult.

Participants reported that flooding had caused changes in the river, leading to the loss of land on the riversides and putting houses and farms close to the river at risk. Some of

the men suggested that they needed to plant forest species such as yacushimbillo, or guava tree (*lnga marginata Willd.*), to reduce soil erosion on riverbanks; however, they did not have sufficient resources.

*3.5. Mental-Health Implications*

Changes in the Shawi food system were also reported to affect Shawi men's well-being and mental health. Within the individual dimension, no longer being able to hunt creates feelings of shame (14 in Figure 4) and guilt (15 in Figure 4), which has increased alcohol consumption (16 in Figure 4) in communities like Nuevo Progreso. During fieldwork, we frequently observed men drinking distilled alcohol at any time of the day. In Nuevo Belén and Nuevo Yurimaguas, Shawi men shared that many of them had decided to turn to evangelism because they needed order in their life to stop abusing alcohol (16 in Figure 4), reflecting the mental-health implications of food-system transitions. Within the community dimension, the perception that their own population growth is the main driver of the reduction of food from the forest is a source of guilt. For Shawi men, the forest is a space that connects the physical with the spiritual world, where people are interdependent with the forest. Any disruption in the plants and animals is often attributed to people, as in this case, where they positioned themselves as the cause of reduced availability of forest food.

Perceptions suggested that changes in forest-food availability are causing an increase in anaemia and malnutrition among the Shawi. Many interviewees reported that they knew about anaemia because they attended a public health centre one hour away on foot. Services at the health centre are only available in Spanish and, according to the interviewees, are not provided respectfully. Iron tablets are given freely to treat anaemia; however, many participants reported that they did not take them. Men recalled that when they visited the public health centre, the health worker told them to hunt more, increasing pressure and burden on the men to hunt despite challenging conditions. Furthermore, rain also interferes with attendance at health centres for nutritional check-ups due to difficult travel conditions. The men explained that treating anaemia at health centres was a long process, and they were given appointments that they were sometimes unable to attend due to heavy rains. They reported that health workers did not recognise the legitimacy of this reason for absence and took it as disrespect or imprudence.

> *"The physician does not recognise reason like weather or increase of water streams. How can we go to the hospital if it's a bad day? When the river is growing, I'm not going to go to the health service, I have to respect the river . . . The doctor wants you to come even when there is rain. He does not understand because in the city it is different. In the city you can go when it is rainy because there are motorbikes, there is public transport but here in the community, you have to walk, so you can't get there, it's very different." –Shawi adult.*

Spirituality is central to the Shawi perceptions of well-being, health, and nutrition. Despite the Peruvian government's promotion of an intercultural approach to healthcare services [57], the Shawi worldview and cultural characteristics are widely considered to be neglected in Peruvian formal health systems. Interviewees expressed a preference for community healers (vegetalistas) to identify the origin of their ill health. The vegetalista then suggests that people visit a healthcare centre when they recognise a disease that they cannot treat effectively. Where an illness is believed to be treated locally, they produce medicine based primarily on local plants, sometimes mixed with paracetamol or other medicine they are familiar with and feel may be beneficial. Additionally, in Nuevo Progreso, people reported seeking the services of shamans for illnesses related to the soul.

## 4. Discussion

Consideration of Indigenous peoples' perspectives of their food system in relation to climate change has shown clear connections between (mal)adaptation strategies and the actions of the Peruvian government, as well as the consequences for their mental health. Our results indicate that Shawi men have been forced to shift from relying on forest foods towards agriculture to provide for their families and maintain their food sovereignty.

In doing so, Shawi men recognise a need for new skills and livelihood strategies for younger generations, expressing hope that government-provided education will provide them and, in doing so, overcome undernutrition and hunger. The desire to obtain new knowledge through public schooling and other public services has driven an increase in Shawi population growth while at the same time contributing to further deforestation in the Amazon, in turn increasing biodiversity loss.

Previous research highlighted that Shawi population growth and desire for education might reflect maladaptive strategies that increase pressures on the forest, increase food insecurity, and constrain adaptation to climate change [45]. However, Shawi's male perspectives expressed in our study indicate that population growth is an intentional response to systematic and historical exclusion perpetuated by the Peruvian government. The government's logic behind service delivery is perceived not as rights-based but situated in inflexible economic efficiency: the worthiness of a community to receive educational services is based on the number of pupils, regardless of differential access and unique geographical or cultural contexts. Despite Shawi men acknowledging that a growing population is detrimental to forest and biodiversity conservation, it is considered to be a necessary step in their strategy if they are to address their current food scarcity, creating a circle of guilt and shame affecting Shawi men's well-being and mental health. Shawi male participants provided a vivid conceptualisation of Indigenous knowledge and strategies that are dynamic, adaptable to change, practical, and often reflecting conflicting priorities.

The analysis and consideration of Indigenous peoples' perceptions within climate-change research draw on growing epistemological calls to take into account the diverse ways in which people and institutions observe, understand, value, and respond to climate change [58–61]. This is important for navigating conflicting development perspectives between communities and policymakers. Our results show that government programs have been focused on producing food to sell at the market (i.e., cocoa), conditional cash transfers, or food-aid programs based on external food sources to reduce food insecurity in Indigenous communities. However, a better way to address these issues might be to change how services are delivered in rural communities. The government´s development and adaptation strategies are embedded in capitalist and neoliberal approaches, which in many cases are unsuited to those who perceive the world differently [58]. Converging these different value systems with an intercultural approach should be a priority in food and climate policy. Consideration of pluralism and acknowledging other worldviews should therefore underpin the conceptualisation of what is meaningful for people and worth being preserved, their envisioned futures, and the best strategies to achieve these futures [29,30,58].

Attention to mental health is needed, in particular given the close relation Indigenous peoples have with place and how rapid socio-ecological changes are causing compounded emotional trauma that reflects—and emerges from—the Shawi worldview. Nu´paru´te is the space where the Shawi people live in harmony with plants, animals, and forest spirits [50]. Consequently, any ecosystem and biodiversity disruption is often associated with their own actions, generating feelings of guilt. Acknowledging the role that worldviews play in driving mental-health outcomes is critical to understanding Shawi food transitions. New research has called attention to the impacts of climate change on mental health, especially emotional responses related to ecosystem loss, which have been conceptualised in the literature as "ecological grief" [31]. This loss is linked to the value people place on the things they are dispossessed of and for which there can be no substitute in the future [29]. This is particularly important in Indigenous contexts such as that of the Shawi because Indigenous peoples often perceive nature as sentient. The emotional trauma expressed by Shawi men in our study arising from the loss of forest food is a manifestation of their worldview. This reflects the grief associated with anticipated future losses of species, their knowledge of accessing food, and their role in the Shawi's traditional social and cultural practices around food, as well as their identity.

Consideration of Indigenous peoples' worldviews also broadens research attention to the spiritual dimension of the forest by reinforcing interdependency relationships: the Shawi depend on the forest for survival, and the forest depends on Shawi traditions to be preserved. Similarly, Indigenous peoples in Ecuador have demanded an understanding of the Amazon as Kawsak Sacha, or living forest. This concept challenges the perception of the Amazon as uninhabited and untapped, full of resources open for exploitation, and of Indigenous peoples as guardians of "an imagined and mythical space, but as reproducers of human and non-human relations and defenders of a territory that is crucial for their lives" [62]. Concepts like Kawsak Sacha or Nu´paru´te question the mitigation of climate-change policies by calling for a reconceptualisation and broadening of their understanding of the Amazon.

## 5. Conclusions and Recommendations

This study highlights the importance of ecosystem conservation for addressing Indigenous food insecurity and climate change in biodiversity hotspots like the Amazon. By understanding Shawi men's perceptions of their food system and how important the forest is for their well-being, we conclude with recommendations for effective adaptation strategies. First, there is a need to re-evaluate the way the Peruvian government is approaching service delivery in Indigenous Amazonian communities, particularly the extent to which it may be increasing deforestation and contradicting efforts to mitigate climate change by indirectly promoting population growth. Second, adaptation initiatives should prioritise agriculture, as Shawi men are actively transitioning towards livelihood dependence on agriculture. However, this activity leaves them highly vulnerable to climate change. Finally, the mental-health implications of current food-system changes and biodiversity loss in Indigenous peoples require urgent attention.

This study also highlights priorities for further research moving forward. First, given the emphasis placed on education by Shawi men as a key component of their envisioned future, further research could evaluate how public schooling is responding to Indigenous peoples' expectations vis a vis food insecurity and climate change. Second, research targeting Indigenous women and youth is critical to complement the presented results. Finally, interdisciplinary research is needed that complements the narratives of Shawi men with an assessment of competition-resource tenure, distribution systems, and foraging behaviours in the past and the present versus agricultural work.

**Author Contributions:** Conceptualization, I.A.-R. and L.B.-F.; methodology, I.A.-R. and C.Z.-C.; writing—original draft preparation, I.A.-R.; writing—review and editing, L.B.-F., C.Z.-C., J.D.F. and P.C.; supervision, L.B.-F. All authors have read and agreed to the published version of the manuscript.

**Funding:** This research was supported by the Indigenous Health Adaptation to Climate Change Program through funding from the Canadian Institutes of Health Research (106372-003, 004, 005). I.A.-R. and L.B.-F. were also supported by funding from a Royal Society Wolfson Research Merit Award (WM160093). C.Z.-C. was supported by the National Institute for Health Research using the UK's Official Development Assistance funding and by Wellcome (218743/Z/19/Z) under the NIHR–Wellcome Partnership for Global Health Research.

**Institutional Review Board Statement:** Ethical considerations for the study were approved by the Institutional Review Board (or Ethics Committee) of the UNIVERSITY OF LEEDS (AREA 18-068, 15 January 2019) and CAYETANO HEREDIA PERUVIAN UNIVERSITY (103420, 21 December 2018).

**Informed Consent Statement:** Informed consent was obtained from all subjects involved in the study. Written and oral informed consent were obtained from the participants and the Nuevo Progreso community.

**Data Availability Statement:** The datasets generated and analysed during the current study are not publicly available due their sensitive nature, but an anonymised version of the interviews (in Spanish) are available from the corresponding author on reasonable request.



**Acknowledgments:** We would like to thank all the people in the Nuevo Progreso community for hosting I.A.-R. and collaborating in this research. Special thanks to Shimer for the translation, to Lea Berrang-Ford for mentoring I.A.-R. in the entire process in which this paper was drafted, to Paul Castro for providing Figure 2, to Alberto Chanchari for Figure 3, and to Tiana Bressan and Ximena Armendariz for proofreading this manuscript.

**Conflicts of Interest:** The authors declare no conflict of interest. The funders had no role in the design of the study; in the collection, analyses, or interpretation of data; in the writing of the manuscript; or in the decision to publish the results.

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
