# Peer review of "Indigenous Peoples’ Perceptions of Their Food System in the Context of Climate Change: A Case Study of Shawi Men in the Peruvian Amazon"

_sustainability, doi:10.3390/su142416502_

Round 1
Reviewer 1 Report
Dear authors,
I read your submission with great interest. Research on sustainability transitions has primarily focused on the liberalization of environmental, social, technological, and institutional dimensions regurgitating Western values. Therefore, solutions call for embracing "unheard voices" and integrating, amongst others, ecological, feminist, and indigenous thinking to address inequality, injustice, and unethical transitions. Therefore, your article addresses a relevant and timely topic and potentially can be of great added value. However, as my role is to provide some constructive criticism, there are certain aspects of your manuscript that you should improve while revising your work. Therefore, I am organizing my review following your submission.
Abstract:
From the abstract, it is unclear what exactly this article is about. What is the exact problem you are investigating, and why? Why would the readers continue reading? The purpose of an abstract is to serve as a "hook" to entice the reader to read your article. Who would be the audience for your article: scholars, practitioners, or both? Construct your abstract to attract your audience.
Introduction:
The introduction tries to establish the research context, which refers to the negative impacts of climate change on the amazon and related disturbances in (indigenous) food systems. More specifically, it contributes to the limited knowledge of the social and cultural aspects of these changes and how indigenous peoples perceive those. However, the quality of the text makes it very hard to judge whether that is precisely their argument. The authors introduce multiple terms, and it is not clear how those are connected. For instance, they talk about Indigenous food systems (IFS), and then about IFS transition.
Furthermore, the authors talk about climate change impacts and drivers, and it remains unclear how those are connected to the social and cultural aspects that this research investigates. Then, at another moment, the text mentions " health-food-climate intersection" without explaining why that is necessary. In other words, the text jumps back and forth without telling the reader a story. The lack of coherency due to the absence of a clear message is also seen in the Discussion section.
A significant overhaul of the introduction is needed to guarantee the logical concisely and structure of the text. For instance, the article could start with the overarching issue - effects of climate change in the Amazon and the associated anthropogenic changes and how those relate disruption of not only local food systems but also lifestyles associated with the arrival of industrialized and neoliberal tools - and continue with the aim, objective and research question, which are currently missing or underdeveloped.
On second thought, the introduction is also where you introduce the overarching context of your research and your unit of analysis. It is only in the methods subsection that the reader is told that this is an article investigating the Shawi men's point of view.
A major letdown for the introduction, and the article in general, is the lack of background materials or a theoretical lens that the authors are taking. At this stage, it feels like data-driven, exploratory research, which is welcome, given the novelty of the topic. First, however, authors should provide background information on the most recent developments in the field and how their research is situated in those.
Materials & methods
The lack of background materials undermines the ideas presented in the "Conceptual framework". How did authors arrive at those and decide to use them?
No major comments for "Study area"
Methods
No major comments. Can the authors clarify the 'male only' participant sample? Presumably, this could relate to cultural or traditional norms, and it is only fair to reveal those to make the methodology more transparent.
Results
The results are presented in a systematic and detailed way. At the same time, it raises questions. The lack of a theoretical or intellectual lens that supports the results does not act in favor of the manuscript. For example, how did the authors arrive at Fig 3, which is central to the results? These dimensions were not outlined explicitly earlier in the manuscript, and it is hard to decide whether they are analytically salient. Authors are encouraged to revise the first part of the manuscript in such a way that it makes the presence of the information presented in the Results section valuable.
Regarding the dimensions in Figure 3, the authors refer to them as rather categorical and the connections between them as one-directional. The text, however, implies that there are no strict boundaries there, and changes in the "individual" bubble could also be coupled with the "societal" or "climatic" dimension, for instance, deforestation and flooding. Perhaps it is the arrows in the diagram. Authors should reflect on their findings and select the essential elements to convey their story. For instance, as enjoyable as it is to read how alcoholism and evangelism are related, it remains unclear how they relate to climate change. The reviewer realizes that all data is valuable, but sometimes, we need to "trim the fat" to deliver a clear and coherent story with our articles.
One minor note, in the "societal" context, the authors mainly describe institutional and neoliberal ideas of development, i.e., new infrastructures, promotion of monoculture, and urbanization. The same applies to the "Health-food-climate intersection" where authors describe lifestyle change choices and conflicts with service providers, who seem to be detached and uninvolved in local customs and IPS beliefs. Perhaps it is a good idea to rethink the function and title of these sub-sections.
Overall, the results are well written, and the advice here is to provide descriptive information regarding culturally relevant issues unfamiliar to the reader. Of course, there are times when the authors do that, but at times one needs to google search to find out what the words in italics mean.
Discussion
The most comment elements worthy of a Discussion section are present. Yet, this reviewer wishes that the authors would have situated the following sentence that opens the section at the beginning of the manuscript:
"We focus on the value-based perspectives and pragmatic decision-making of Shawi men to highlight current and potential adaptation strategies and trade-offs related to the health-food-climate intersection " (page 12)
Doing so will not only clarify what the article is about but also provide a cornerstone around which the manuscript can be built upon. However, introducing it in the Discussion is, as they say, "too little, too late". A second issue here also refers to the "health-food-climate intersection," which needs further explanation. The results indeed touch upon the topic but remain short of logical reasoning. Suppose that was one of the central concepts used in the manuscript. In that case, it should be dissected convincingly and positioned within the relevant literature and the study context, not in the Discussion.
At times, new information or references are included in the Discussion, which should have been introduced earlier. The function of a discussion is mainly to summarize your key findings and give your interpretations of these. While you do that successfully, you should rely on sources you used earlier rather than introducing new ideas and information. For example, mental health was not mentioned anywhere earlier. Again, this could be easily fixed if authors had a background section where they elaborate on their ideas. That being said, the implications and recommendations are well-written and to the point.
One minor note here is that authors could be a little bit critical or perhaps bolder in their language. For example, there is a fine line between IPS, as you write now, "consciously considering transitioning from reliance on forest foods towards farming and breeding" and "being resorted" or "forced" to transition. Namely, the thick data authors have aggerated supports this type of speculative statement and highlights some of the challenges IPs face.
Conclusion
The conclusions are succinct and repeat some of the points made in the Discussion. This problem arises mainly because the manuscript tackled no straightforward research question(s). Therefore, in rewriting the Conclusions, the authors are also advised to reflect on restating the aims of the study, explaining the significance of the findings or contribution of the research, and acknowledging limitation(s) while stating a result or contribution.
Minor comments thought all the manuscript: Extensive editing of the English language and style is required
Reviewer 3 Report
Great topic. Congrats.
I have a few recommendations:
1. In the Abstract, lines 17 and 22, you repeat Our results. Try to change the expression to not have a repetition there
2. Line 24. There is a grammatical error: Probably is are creating? not are create...? I wouldn't put the word however between are and creating though.
3. The abstract should be shorter. I would not present in the abstract the conclusion. Lines 27-30
4. You must add a Literature review section and in it you should add more references. Also check the guidelines for the references in the text, you put them in brackets, like [13-15], [13,15]. Check the entire paper
5. Figure 1. There you should put a source for the map
6. Figure 2. Congratulations for that!! and for the person who draw that. A really original addition to the paper.
7. Line 313. Do not put the full stop before the brackets. And use Figure not fig. Do the same for the others too.
8. Figure 4. You should have a written agreement from their parents regarding the picture. It is about GDPR and privacy and children's protection. I am not against it per se.
9. It was not clear for me why only males were interviewed. Maybe you could explain it better in the paper. And how choosing only men affected or not the research in your opinion (you can put that in Limits of the research in conclusion)
10. The conclusions> Add short paragraphs to present theoretical and practical implications of your paper, limits of your research and future research directions.
9. Add more recent references (2020-2022) and check te guidelines for formatting them
10. In figure 1, there are Spanish words I think. Is there an English version of the map? also resolucion in the centre of the image.... Just a recommendation
Concluding, I really congratulate for the topic and for spreading news about the community there. I think this research is meaningful!!!
Round 2
Reviewer 1 Report
Dear Authors,
After reviewing your revised manuscript and reading the accompanied file with your feedback on my feedback, I feel you have done an excellent job in improving the manuscript.
While I have no major comments, there might be two minor issues that you should look at:
page 4, lines 128 - 131, would be ideal if authors mention the time frame of the survey.
Figure 5 - Please check with the publisher / journal whether obscuring of faces for non-identification of the underage person depicted on the photo is required
Thank you for working on such an important and underrepresented research area.
Author Response
Dear Reviewer 1,
Thank you so much for your comments, we really appreciate you enjoyed reviewing the paper. Another English round by each author has been made. Please find a detailed answer to your comments:
- Page 4, lines 128 – 131: We added the time frame of the survey (L128-130)
“Based on a household questionnaire in 2014, a study found that food insecurity and malnutrition levels among the Shawi were among the highest globally: 98% of Shawi households were food insecure, 66% of children had anaemia, 44% were stunted, 17% were underweight, and 19% of anaemic children had an overweight parent [22].”
- Figure 5: We have been in contact with the editorial board about the picture and now the face is blurred.
Kind regards,
Reviewer 2 Report
Dear Editor,
Some minor revisions are needed.
I have attached the file here with it.
I think, this study is already done but the explanation and intext writing seems present form or need to be done.
I asked to the author to check thoroughly the entire manuscript again with the reference style.
You can accept this manuscript in present form.

Author Response
Dear Reviewer 2,
Thank you so much for your comments. All co-authors have done another test reading in English. Please find a detailed answer to your comments:
|
Reviewer’s comments |
Our response |
|
L85: I think this study is already done. Please check the English again. |
A round of English edits has been made again to check past writing. |
|
L120: Delete it. Write like: Source adopted from Pörtner et al. 2022 [1] and modified by the authors. |
Thank you. We have changed as you suggested. |
|
L146: Add proper source and software. |
We have added the source and software: “Map outlining the study area made by Paul Castro for this manuscript with ArcGis 10.8.1 Software. Source: Official database from the Agriculture Ministry of the Peruvian Government (Community limits), National Institute of Statistics and Informatics (town locations), Ministry of transport and communications (road).” |
|
L236: Source: Pictures taken by Matthew King for the IHACC program. |
L236: Added: “Source: pictures taken by Matthew King for the IHACC program.” |
|
L265: Italic in (Cripturellus cinereus). |
L265: Done, now in L269. |
|
L330: Check upper case and lower case. |
L330: Changed, as well as the other subtitles to be consistent L383, L452. |
|
L365: Source: This picture is shared with the permission of Alex and his parents Olegario and Sandra. |
L365: added. |
Kind regards,
Reviewer 3 Report
Congratulations on your work. I hope this type of work and research will continue. It is really important and also what I most liked is the mix between research and the fact that it is also meaningful. I miss that when I read and review papers. Qualitative studies should be encouraged. Congrats again.
Author Response
Dear Reviewer 3,
Thank you so much for your comments. All co-authors have done another proofreading in English.
Kind regards,